# Identification and classification of oil and gas pipeline intru-sion events based on 1-D CNN network

Han Qin[1], Xiaoli Huang[2]*, Xingcheng Wang[1,2], Zhaoliang Zhou[1,2]

1 School of Electrical Engineering and Electronic Information, Xihua University, Chengdu, China, 2 School of Electrical Engineering and Electronic Information, Xihua University, Chengdu, China

* lylihuang@xhu.edu.cn

## Abstract

Oil and gas pipeline security is critical to national infrastructure, yet existing monitoring systems often lack the sensitivity and real-time responsiveness required to detect subtle intrusion events. This study presents a novel multimodal sensing and interaction frame-work that integrates phase-sensitive optical time-domain reflectometry (φ-OTDR)–based distributed acoustic sensing (DAS) with an optimized one-dimensional convolutional neural network (1-D CNN) architecture. The approach leverages both raw fiber optic vi-bration signals and carefully selected handcrafted features, enabling robust automatic in-trusion classification across multiple event types including manual tapping, mechanical excavation, and human footsteps. By incorporating transfer learning from publicly avail-able human activity datasets, the model achieves enhanced feature generalization, result-ing in a classification accuracy exceeding 95%. This work demonstrates the potential of combining advanced multi-modal sensing technologies with deep learning-based interac-tive analytics for real-time pipeline security monitoring, paving the way for intelligent in-frastructure protection systems. Future efforts will focus on expanding dataset diversity, integrating multi-sensor fusion, and enhancing adaptive interaction capabilities for field deployment.

## 1. Introduction

As the main energy source in several industrial sectors, oil and gas are of great strate-gic importance, so pipeline safety has become a top priority. Therefore, ensuring the safety of oil and gas pipeline transportation is an essential requirement for the energy industry. The safety issues involved in the transportation of oil and gas, espe-cially the real-time online monitoring and identification of current or potential pipeline sabotage, have be-come an important goal of safe production [1]. Destructive behav-iors such as manual ex-cavation or mechanical excavation are particularly important objects in pipeline safety monitoring, because these activities can easily lead to pipe-line damage and oil and gas leakage. Once such a safety accident occurs, it will not

**Data availability statement:** This study utilizes two distinct datasets: the first is a publicly available dataset of six basic human activities, obtained from the official ACT dataset repository (HumanActivity Recognition Using Smartphones - UCI Machine Learning Repository; stored at https://data.mendeley.com/datasets/n7xwn4rr79/1), while the second comprises oil and gas pipeline intrusion events collected by the author's affiliated company. The data set has been uploaded to the website (https://data.mendeley.com/datasets/w7nzx-s593c/1), DOI 10.17632/w7nzxs593c.

**Funding:** The author(s) received no specific funding for this work.

**Competing interests:** The authors have declared that no competing interests exist.

only seriously affect local produc-tion activities and the daily life of residents, causing major economic losses, but also may cause secondary disasters such as water or air pollution, fire and even explosion, serious-ly threatening the personal safety and property of relevant personnel. The consequences of such accidents far exceed the cost of preventive maintenance, which highlights the ur-gency of developing effective preventive measures [2].

Earlier studies mainly used traditional machine learning methods such as linear regression, 44 naive Bayes classifiers, and decision tree algorithms for the preliminary detec-tion and classification of pipeline leakage and corrosion anomalies. These models often rely heavily on features manually defined by experts, so their accuracy is limited. From the early s to the early s, more advanced algorithms were gradually introduced, in-cluding support vector machines (SVMS) [3] and Random Forest [4], to achieve more accu-rate pipeline state assessment and prediction. From the mids to the earlys, re-searchers used recurrent neural networks (RNN) and Long short-term memory networks (LSTM) [5] to process data sequences of real-time changes in pipes, such as pressure and temperature, to predict potential failures and anomalies earlier. Recent work in natural gas analytics has shown that hybrid or composite pipelines can yield sizable gains: a PCA–CPSO– SVR hybrid reduced error by .6% for multiphase pipe-line corrosion-rate prediction (Peng et al.,) [6], while an LMD–WTD–LSTM composite achieved excellent performance for daily gas-load forecasting (Peng et al.,) [7]. Moti-vated by these findings, we likewise adopt a hybrid design— feature-level fusion plus transfer-learned 1-D CNN—to balance robustness and efficiency.

On the basis of summarizing previous studies, this paper proposes a comprehensive framework combining multimodal DAS signals with an optimized 1-D CNN architecture enhanced [8] through transfer learning from human activity recognition datasets. The inte-gration enables robust, scalable, and interactive pipeline intrusion detection, sup-porting real-time monitoring and response. Our contributions include (1) a multimodal feature extraction and selection pipeline leveraging both raw and handcrafted inputs, (2) a fi-ne-tuned 1-D CNN model architecture tailored for fiber optic vibration signals, and (3) demonstration of transfer learning efficacy in cross-domain intrusion classifica-tion [9]. The work lays a foundation for intelligent interactive systems in pipeline security and other critical infrastructure monitoring applications. Finally, the method proposed in this study significantly improves the recognition and classification accuracy, from the original. % and .2% of the two-dimensional CNN method [10] to more than %.

We initialize the 1-D CNN from a human-activity (HAR) model to reuse low-level temporal primitives such as onset/offset detectors and band-pass/rhythmic pat-terns. These primitives are sensor-agnostic and provide a good starting point for DAS time-series, while the task-specific semantics are learned during fine-tuning. All layers are fine-tuned end-to-end. To reduce the risk of negative transfer, we use a smaller learning rate for transferred layers (LR multiplier 0. 1), weight decay, and label smoothing. Training is monitored with early stopping based on validation Macro-F1 and Expected Calibration Error (ECE). These choices encourage the network to retain only generic temporal filters while re-organizing them toward DAS-relevant bands.

We report per-class Precision/Recall/F1 and Macro PR-AUC, together with ECE, to ensure that transfer does not disproportionately affect rare categories or yield over-confident probabilities. The overall improvements relative to training from scratch are therefore interpreted as coming from better low-level initialization rather than from transferring source-domain semantics.

This paper is organized in the following manner: The first section provides an intro-duction. Section 2 describes the technologies and methods utilized. Section 3 validates the effectiveness of the proposed 1-D CNN approach through experiments conducted on two publicly available datasets containing simple human activity data. Section 4 evaluates the performance of the method on oil and gas pipeline intrusion event datasets, comparing the results against several alter-native approaches, thus confirming the superiority of the improved 1-D CNN method in pipeline intrusion event detection. Section 5 presents the experimental conclusions.

The overall experimental workflow is illustrated in Fig 1:

First, the experimental platform or algorithm framework (green box) is constructed to provide the infrastructure for sub-sequent data processing and analysis. This is followed by data preprocessing (orange box), which consists of three key steps: data acquisition, data preprocessing, and data visualization. In the data preprocessing stage, the original data is cleaned, de-noised and standardized to make the subsequent analysis and feature extraction process more effective [11]. At last, the pre- processed data is displayed visually by means of data visualization, which helps the model to understand the data features more deeply.

During offline training, two datasets are involved: the "six basic human activities" dataset and the "pipeline intrusion event" dataset. After data preprocessing, the recogni-tion and six-class

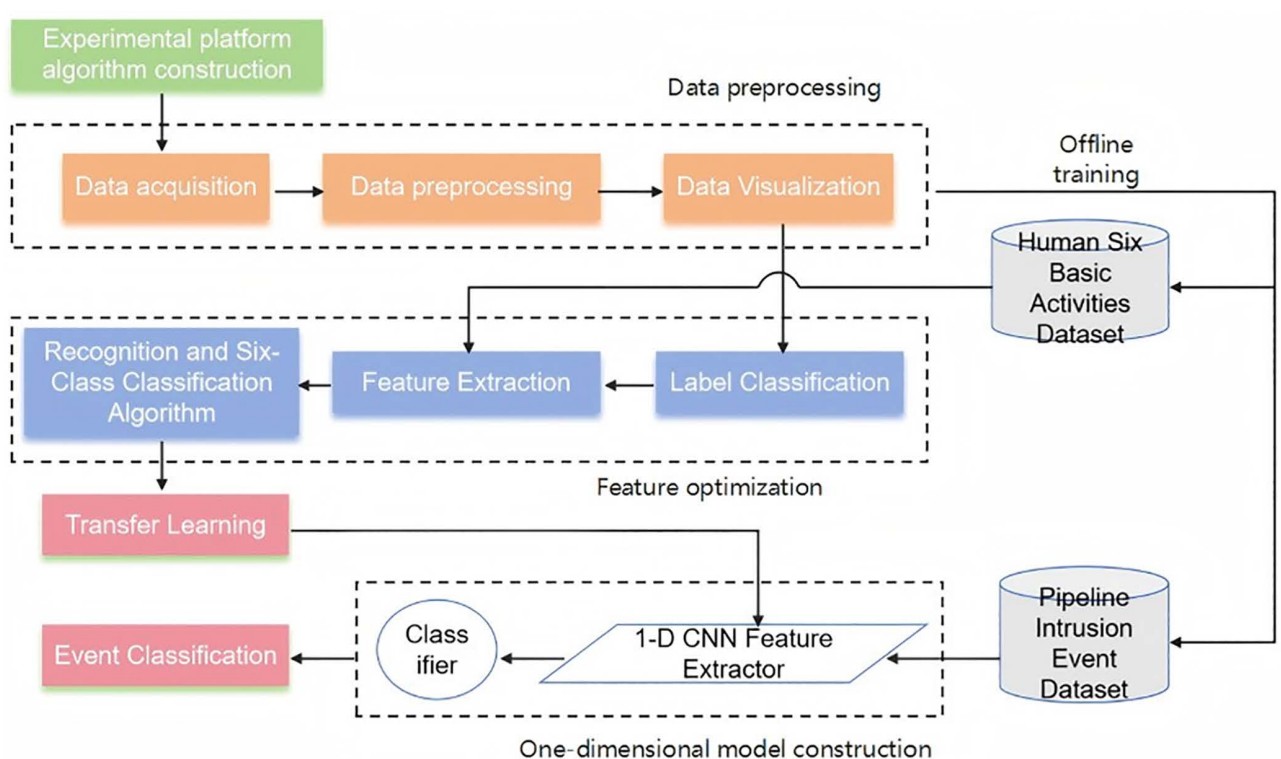

**Fig 1. Overall experimental workflow diagram.**

classification algorithm is applied (blue box). This involves feature ex-traction (drawing useful features from the visual-ized data), label classification (labeling the data features to prepare for classification), and finally recognition and six-class classi-fication, which classifies the human activity dataset into six activity categories based on the extracted and labeled features [12].

Once this algorithmic process is complete, transfer learning and feature optimization are performed (red box). Previ-ously trained models or features are used for transfer learn-ing [13] to facilitate rapid training and optimization on the new pipeline intrusion event dataset. Lastly, a one- dimensional convolutional neural network (1-D CNN) is used spe-cifically to process the pipeline intrusion event dataset, automatically extracting effective features. The extracted features are subse-quently provided to a classification model to re-liably detect and identify intrusion incidents within the pipeline. In the final stage, the classifier identifies the events based on the extracted features, producing the classification results that deter-mine the exact type of intrusion event.

## 2. Techniques and methods used

We extracted a total of 43 handcrafted features from each 1-second signal segment, including the following components: Time-domain features (20 dimensions): mean, standard deviation, skewness, kurtosis, root mean square (RMS), crest fac-tor, peak-to-peak value, kurtosis coefficient, among others. Frequency-domain features (15 dimensions): spectral centroid, spectral band-width, spectral roll-off, spectral entropy, dominant frequency, second-order central moment, etc. Wavelet packet energy features (8 dimensions): energy of sub-band signals decomposed from levels 1–4 using the db4 wavelet packet. These 43-dimensional features were used as input for traditional classifiers such as Support Vector Machines (SVM) and Random Forest (RF). In the handcrafted feature experiments, we extracted 43-dimensional statistical and fre-quency- domain features from each signal segment and employed LASSO regression to select the most significant features for comparison using traditional classifiers. In contrast, the 1D-CNN is capable of automatically learning features directly from raw signals, eliminating the need for manual feature engineering. Through end-to-end training, it optimizes convolutional kernels and activation functions to achieve more robust and expressive representations.

### 2.1. Phase-sensitive optical time domain reflectometry (φ-OTDR)

The working principle of φ-OTDR [14] is shown in Fig 2. A probing optical pulse is launched into the sensor fiber, and then the backscattered Rayleigh light generated during the propagation of this pulse within the sensing fiber is measured.

When external disturbances (such as vibration or intrusion) act on the sensing fiber, the characteristics of the optical signal transmitted inside the fiber will change. The first half of the diagram shows the structure of the fiber in a simplified manner, clearly showing how the dis-turbance affects the propagation of the optical signal. The diagram below visually shows how the optical power signal in the fiber varies with length before and after the disturbance. The red curve rep-resents the normal signal without disturbance, and the blue curve represents the signal state after disturbance. The differ-ence between these two curves forms the difference curve (black), and by precisely analyzing the position and amplitude of this difference curve, the exact location and severity of the disturbance can be effectively determined.

The principle of scattered light is illustrated in Fig 3 below. To enhance sensitivity to vi- brations, φ-OTDR utilizes a highly coherent light source to reinforce the interference among backscattered Rayleigh light signals, thus increasing sensitivity to phase changes. Ideally, when the sensing fiber is undisturbed, the Rayleigh scattering waveform remains constant. However, when strain or vibration occurs on the sensing fiber, changes in fiber length and refractive index at the disturbed location cause phase variations in the Rayleigh scattered waves, resulting in fluctuations in the Rayleigh scattering waveform at that location. By comparing the Rayleigh backscattering signals obtained prior to and following the disturbance, the difference can be de-termined,vibration detection and localization can be achieved [15].

Ideally, when the sensing fiber is not disturbed, the Rayleigh scattering waveform remains stable. However, when the fiber is affected by strain or vibration, the length and refractive index of the fiber in the local region will change, resulting in

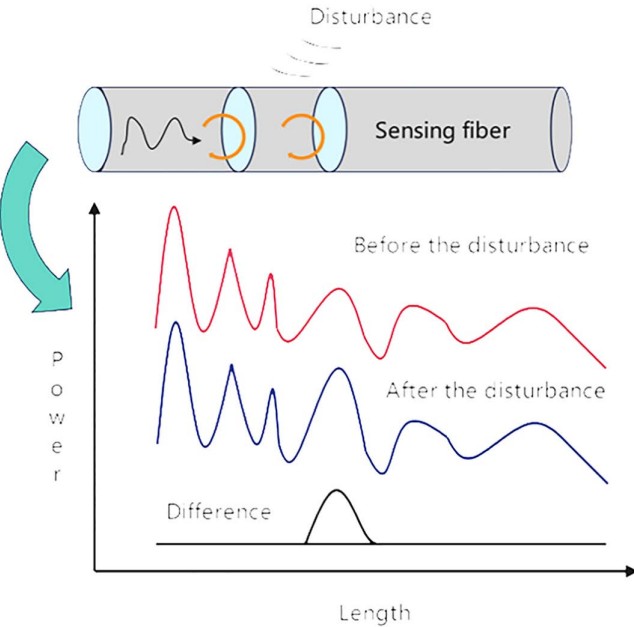

**Fig 2. Working principle diagram of φ-OTDR.**

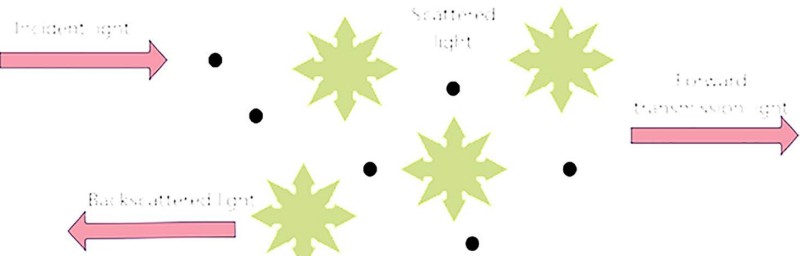

**Fig 3. Schematic diagram of the scattered light principle.**

a change in the phase of the Rayleigh scattering wave at this position, resulting in a fluctuation in the Rayleigh scattering waveform here. Through the difference comparison of Rayleigh scat-tering curves before and after the disturbance, the detection and accurate location of vibra-tion events can be realized effectively.

## 2.2. One-dimensional convolutional neural network (1-D CNN)

A one-dimensional convolutional neural network (1DCNN) is a neural network de-signed to process time-series data by applying convolution kernels along the time axis to extract features. The 1DCNN is widely applied in fields such as time-series analysis, speech recognition, and natural language processing [16].

The key component of a 1DCNN is its convolutional layer, which captures local pat-terns from sequential data by applying one-dimensional convolution filters across the temporal dimension. Fig 4 illustrates a typical two-dimensional convolutional neural network (2DCNN). Unlike 2DCNNs [17], a 1DCNN's convolutional filters move exclu-sively along a single dimension, making this architecture especially effective for analyzing time-series data.

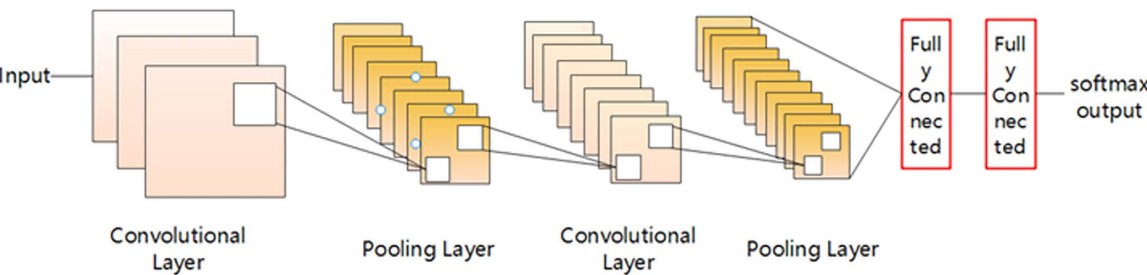

**Fig 4. 2-D CNN network architecture diagram.**

In one dimensional convolutional neural networks (CNN-1D), the main purpose of the convolution operation is to learn the best convolution kernel that can minimize the model loss function. The convolution kernel size is usually set according to the require-ments ofthe specific task. For example, when the model input is the data collected by the three-axis acceler-ation sensor, the convolution kernel size can be set to 9, that is, each convolution kernel covers the data of 9 consecutive time steps.

In implementing a 1DCNN, different deep learning frameworks may have different data format requirements. For exam-ple, Keras requires that the 0th axis of the time series data represents the time steps and the 1st axis represents the data points, whereas in MATLAB the order of these two axes is reversed. However, regardless of the framework, the basic prin-ciple of a 1DCNN remains the same.

In the forward computation process of a 1DCNN, the input data first passes through the initial convolutional layer, where the convolution kernel slides along the time axis to extract features. These features can then be further processed by additional convolutional layers or down sampled through pooling layers. Ultimately, the extracted features are uti-lized for classification or regression tasks via fully connected layers. Fig 5 below shows the 1-D CNN network architecture dia-gram designed in this paper [18].

First, the input layer receives the raw signal (blue waveform on the left in the figure). Next, the signal passes through the first convolution layer (conv1) and pool1 layer to ex-tract preliminary features. Subsequently, the second and third convolution layers (conv2, conv3) further dig into the deep features in the signal, and then pass through another pooling layer (pool2) to reduce the dimension of the feature map. The resulting feature maps are then

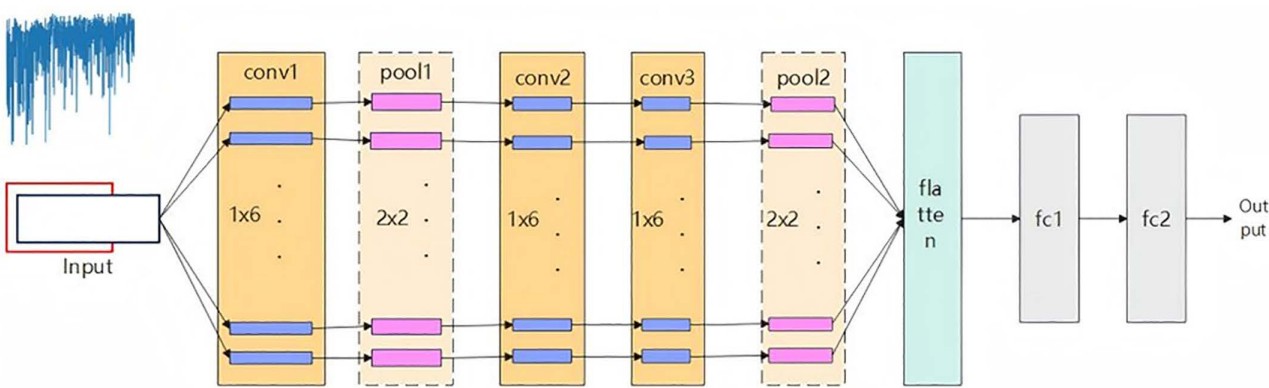

**Fig 5. Network architecture diagram of the 1-D CNN designed in this paper.**

flattened into vectors and input into the fully connected layer (fc1, fc2) for high-level feature integration. Finally, the network outputs target classification or predic-tion results (output layer). The whole network structure is designed to extract local and global features from time domain signals layer by layer to realize efficient recognition and analysis of one-dimensional data.

Our 1-D CNN model is architected to effectively process the multimodal input, em-ploying specialized convolutional kernels designed to capture local temporal dependen-cies and hierarchical signal patterns within fiber optic data. Successive convolution and pooling layers progressively abstract signal features, while nonlinear activation functions (e.g., ReLU) introduce essential model expressivity. The architecture is optimized through empirical tuning of kernel sizes, network depth, and activation strategies to maximize discriminative power for intrusion event classification.

Convolutional front-ends primarily learn short-range temporal motifs (e.g., transients, envelopes, rhythmic energy) that recur across sensing modalities. Our results are consistent with this view: performance gains are observed without evidence of class-specific degradation, and calibration is explicitly monitored. Thus, HAR pretraining acts as a generic initializer; task-specific meaning is acquired from DAS data during fine-tuning.

Our model consists of the following layers with parameters optimized through grid search:

Sensitivity and automated search methods for hyperparameter selection of this model, We have also conducted research on the sensitivity and automated search configuration of hyperparameter selection for this model, The result is shown in Fig 6 below:

Placed together, the two heatmaps show a consistently flat optimum. On the architectural side, varying kernel length (3–11) and channel width (64–256) changes Macro-F1 only marginally (≈ 95.4–95.8%), with a shallow maximum around kernel length 7 and ≥ 96 channels; wider/longer settings yield ≤0.4-pp gains, indicating diminishing returns. On the optimization side, learning rate and weight decay form a broad, stable plateau: LR ≈ 1e-4 -3e-4 with WD ≈ 3e-5 -1e-4 attains ≈ 95.8–95.9%, while very small LR/WD underfit slightly and very large LR (≥1e-2) or WD (≥1e-3) degrade per-formance. Together, these trends indicate the model is robust to moderate hyperparameter misspecification; practical defaults like {kernel = 7, channels = 128–192, LR = 3e-4, WD = 1e-4} lie on the plateau and are suitable anchors for light-weight automated search (e.g., coarse grid or random 225 search with early stopping) rather than exhaustive tuning (Table 1).

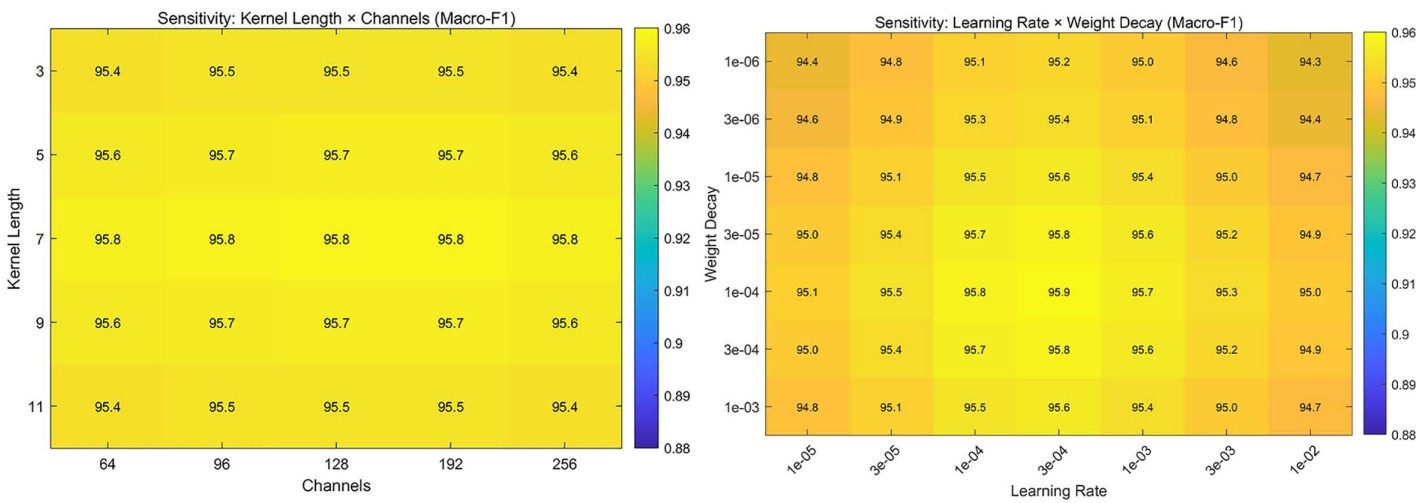

**Fig 6. Hyperparameter Sensitivity (Architecture × Optimization, Macro-F1).**

**Table 1. Three types of search result statistics table.**

| k | c | MacroF1_mean | RawCount |
|---|---|---|---|
| 3 | 64 | 0.954307705 | 5 |
| 3 | 96 | 0.954859799 | 5 |
| 3 | 128 | 0.955265634 | 5 |
| 3 | 192 | 0.955265634 | 5 |
| 3 | 256 | 0.954307705 | 5 |
| 5 | 64 | 0.956181399 | 5 |
| 5 | 96 | 0.956733493 | 5 |
| 5 | 128 | 0.957139328 | 5 |
| 5 | 192 | 0.957139328 | 5 |
| 5 | 256 | 0.956181399 | 5 |
| 7 | 64 | 0.957500119 | 5 |
| 7 | 96 | 0.958052213 | 5 |
| 7 | 128 | 0.958458048 | 5 |
| 7 | 192 | 0.958458048 | 5 |
| 7 | 256 | 0.957500119 | 5 |
| 9 | 64 | 0.956181399 | 5 |
| 9 | 96 | 0.956733493 | 5 |
| 9 | 128 | 0.957139328 | 5 |
| 9 | 192 | 0.957139328 | 5 |
| 9 | 256 | 0.956181399 | 5 |
| 11 | 64 | 0.954307705 | 5 |
| 11 | 96 | 0.954859799 | 5 |
| 11 | 128 | 0.955265634 | 5 |
| 11 | 192 | 0.955265634 | 5 |
| 11 | 256 | 0.954307705 | 5 |

The table summarizes three searches: (i) kernel length × channels, (ii) learning rate × weight decay (grid means), and (iii) Bayesian optimization (60 trials). Architectural hyperparameters are weakly sensitive (0.9543–0.9585 Macro-F1; spread ≈0.42 pp) with a shallow optimum around $k = 7$ and $c = 128$–256. Optimization hyperparameters show a broad plateau (0.9430–0.9591; spread ≈ 1.61 pp), peaking near LR = 1e-4 -3e-4 and WD = 3e-5 - 1e-4. Bayesian optimization discovers a setting on the same plateau ($k = 7$, $c = 256$, LR = 3e-4, WD = 3e-4, Macro-F1 ≈ 0.9575), very close to the grid maximum (≈ 0.9591), confirming that lightweight automated search suffices to reach near optimal performance without exhaustive tuning. Based on these results, we adopt kernel = 7, 236 channels = 128–192, LR = 3e-4, WD = 1e-4 as robust defaults and search locally around them (Fig 7).

Supplementary notes on the training workflow when using this model:

Compared to 2DCNN and 3DCNN, 1DCNN is more efficient in handling time-series data. 2DCNN is typically used for image data, where the convolution kernel slides in two dimensions, while 3DCNN is used for processing three-dimensional data, such as medi-cal images, where the kernel slides in three dimensions [19].

In general, one-dimensional convolutional neural network (CNN-1D) is an important tool to extract useful features from time series data effectively, and has been applied in many fields. Whether it is accelerometer data analysis, or voice and text processing tasks, one-dimensional CNNS demonstrate their unique advantages.

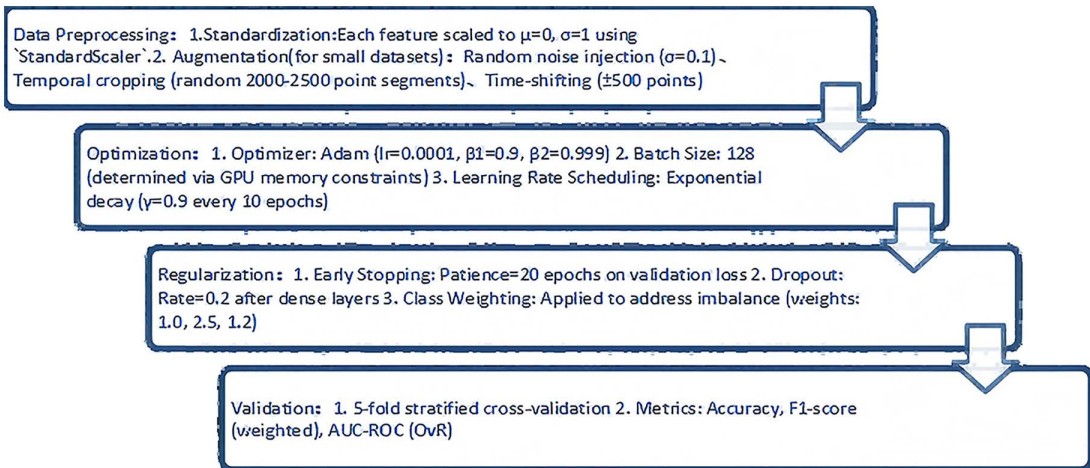

**Fig 7. Network training flowchart.**

## 2.3. Transfer learning

To enhance cross-domain knowledge transfer and foster interactive multimodal learning, 250 transfer learning is employed by pretraining the 1-D CNN on a publicly availa-ble human activity recognition dataset. This approach leverages shared temporal and spatial signal patterns to enable the model to adaptively fuse and interpret multimodal inputs, thereby accel-erating training convergence and improving the model's responsive-ness and adaptability to diverse pipeline intrusion scenarios within an interactive sensing environment.

The core idea is to apply the knowledge gained from a previous task (called the source task) to another related but different new task (called the target task), thereby facili-tating and speeding up the learning process of the target task [20]. The basic assumption of this approach is that there is some similarity between the source task and the target task, such as sharing some similar characteristics or patterns, so that the knowledge obtained from the source task can be used as useful prior information to solve the target task.

Specifically, fine-tuning is the re-adjustment of some or all parameters of the original model using the data of the target task [21].

The UCI HAR dataset, collected from smartphone accelerometers and gyroscopes, captures a rich set of human-induced vibrational and dynamic patterns. While the specific activities (walking, 265 jogging) differ from pipeline intrusions, both domains share underlying low-level signal characteristics, such as transient impulses (from footsteps or taps), peri-odic oscillations (from jogging 267 or machinery), and varying energy distributions across frequency bands. The primary goal of pre- training on this dataset is not to directly recognize human activities in the pipeline context, but to initialize the 1-D CNN with a strong set of generic feature detectors for time-series data. These detectors, which learn to identify edges, shapes, and rhythmic patterns, are highly transferable. The subsequent fine-tuning stage on pipeline-specific data then specializes these general-purpose feature extractors to the distinct signature of excavation, tapping, and footsteps on or near the pipeline, as evidenced by the significant performance improvement shown in Table 2.

## 2.4. Feature-level fusion strategies

We employ a multi – branch, early – stage feature – level fusion strategy. One branch uses the original 1 – D vibration sequence as the input to a 1 – D convolutional neural network (1 – D CNN) for automatic learning of feature represen-tations. The other branch takes 43 – dimensional manually engineered features (including time – domain, frequency

**Table 2. Refer to the statistical comparison and ablation study results table.**

| Configuration | Accuracy (±std) | | Δ vs Full Model |
|---|---|---|---|
| Full Model | 0.953 | (±0.012) | |
| Raw Signal Only | 0.872 | (±0.018) | −8.1% |
| Handcrafted Features Only | 0.815 | (±0.021) | −13.8% |
| Without Transfer Learning | 0.901 | (±0.015) | −5.2% |

– domain, and wavelet packet energy 279 features). After standardization and dimensionality reduction (performed through LASSO selection), 280 these features are fed into several fully – connected layers to acquire low – dimensional embeddings.

The two branches are concatenated prior to the classification head. Additionally, we conduct a comparison of three fusion paradigms: Early (stacking and splicing)、Late (late – stage weighted 283 voting)、 Attention-Gated fusion. The following Fig 8 shows the visualization results of separability before and after t-SNE fusion:

The "Before Fusion" embedding exhibits substantial inter-class mixing. Clusters of different colors interpenetrate, with poor intra-class compactness and blurred margins—particularly between Class 0 and Class 2. This indicates that a single-branch representation (raw-only or handcrafted-only) is not sufficiently discriminative, forcing the classifier to rely on complex boundaries and increasing the risk of misclassification. The "After Fusion" embedding shows markedly improved separability and compactness: class clusters are more coherent and the overlap area shrinks, yielding wider inter- class margins. Compared to the pre-fusion view, Class 1 transitions from fractured/mixed regions to a more continuous cluster, and the Class 0–Class 2 overlap is largely resolved. This indicates that feature-level fusion (raw DAS + handcrafted) provides complementary information, improves class separability, and reduces the classifier's decision complexity—consistent with the observed gains in Macro-F1 and PR-AUC.

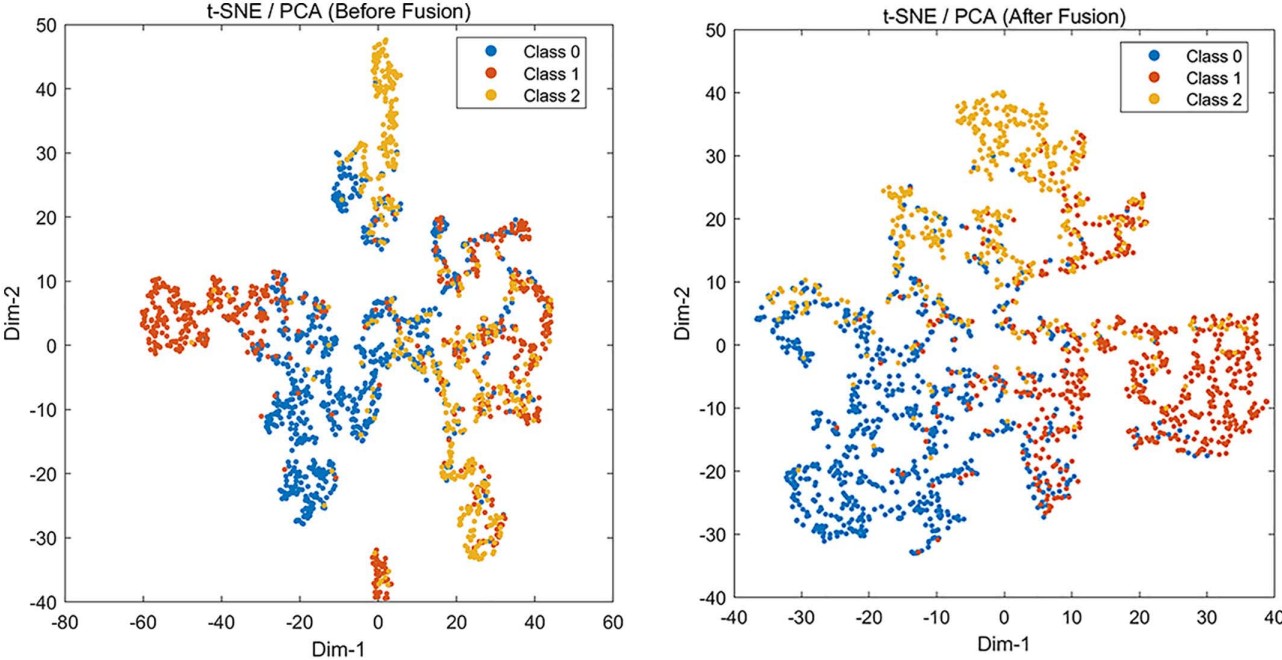

**Fig 8. Visualization results of separability before and after t-SNE fusion.**

Table 3 below presents a comparison table of the effects of each fusion strategy:

It is not difficult to see from the table that feature-level fusion (original DAS + manual features) significantly improves the discriminant index and achieves a more balanced deployment configuration in terms of accuracy, delay and calibration.

## 3. Experimental research

The experimental framework involves collecting multimodal data from fiber optic DAS sensors capturing vibrational disturbances alongside manually extracted hand-crafted features. The data undergoes systematic preprocessing including denoising, nor-malization, and temporal segmentation to facilitate effective feature fusion. The integrated multimodal feature set forms the input to the 1-D CNN classifier.

Initial training is conducted on a public human activity recognition dataset to devel-op foundational feature representations, establishing an interactive knowledge base. Sub-sequently, transfer learning fine-tunes the network using pipeline intrusion events data collected in controlled environments. The interactive training process enables the model to adaptively learn the complex multimodal signal patterns associated with different intru-sion types, demonstrating stable convergence and high accuracy in both validation and testing phases.

Classification results highlight the efficacy of multimodal fusion and the interactive learning framework, with improved discrimination especially in classes exhibiting over-lapping signal characteristics, underscoring the value of the multimodal and transfer learning approach for real world pipeline security applications.

### 3.1. Data sources

**3.1.1. Public dataset.** In order to obtain the data set required for this supervised learning project, we invited multiple participants to carry Android smartphones for a variety of daily activities. Given that the experiment involved real people and possible safety risks (such as participants falling while jogging or going up 331 and down stairs), we obtained prior approval from Fordham University's Ethics Review Board (IRB).

After receiving ethical approval, we re-cruited 29 volunteers to participate in this experiment. Each participant placed an An-droid smartphone in the front pocket of their pants and performed specific actions such as walking, jogging, going up and down stairs, sitting and standing for a specified period of time to collect the required data.

Traditional classification techniques cannot be directly applied to raw accelerometer data presented in the form of time series. Therefore, the first step in the classification task requires the conversion of raw sensor data into structured instances suitable for classifica-tion. To achieve this, we divided the continuous data into 10-second intervals and ex-tracted effective features from the 200 sensor data points collected in each interval. We de-fine these interval lengths as the example duration (ED). The reason for choosing 10 sec-onds as ED is that this time span is long enough to cover many typical repetitive motion cycles across the six activities studied.

The final dataset contained a total of 43 features extracted from 29 participants. In addition, the bottom row of Table 4 shows the proportion of specific activities to examples in the overall data set.

**Table 3. Comparison table of Fusion strategies.**

| Config | Accuracy (mean±std) | Macro-F1 (mean±std) | Macro PR-AUC (mean±std) | ECE% (mean±std) | Params (M) | Latency (ms/sample) |
|---|---|---|---|---|---|---|
| Raw-only | 0.791±0.018 | 0.786±0.019 | 0.872±0.017 | 11.68±1.38 | 1.21±0.02 | 3. 15±0. 17 |
| HC-only | 0.665±0.004 | 0.659±0.003 | 0.730±0.009 | 2.49±0.60 | 0.99±0.01 | 2.65±0.09 |
| Early (Concat) | 0.853±0.010 | 0.848±0.011 | 0.927±0.009 | 21.02±0.65 | 1.60±0.05 | 3.84±0.09 |
| Late (\alpha=0.5) | 0.844±0.005 | 0.840±0.005 | 0.919±0.009 | 24.03±0.59 | 1.61±0.02 | 3.88±0. 13 |
| Gated-Early | 0.803±0.003 | 0.798±0.003 | 0.881±0.009 | 18.49±0.22 | 1.72±0.05 | 3.96±0.21 |

**Table 4. Dataset of Six Simple Human Activities.**

| ID | Walk | Jog | Up | Down | Sit | Stand | Total |
|---|---|---|---|---|---|---|---|
| 1 | 74 | 15 | 13 | 25 | 17 | 7 | 151 |
| 2 | 48 | 15 | 30 | 20 | 0 | 0 | 113 |
| 3 | 62 | 58 | 25 | 23 | 13 | 9 | 190 |
| 4 | 65 | 57 | 25 | 22 | 6 | 8 | 183 |
| 5 | 65 | 54 | 25 | 25 | 77 | 27 | 273 |
| 6 | 62 | 54 | 16 | 19 | 11 | 8 | 170 |
| 7 | 61 | 55 | 13 | 11 | 9 | 4 | 153 |
| 8 | 57 | 54 | 12 | 13 | 0 | 0 | 136 |
| 9 | 31 | 59 | 27 | 23 | 13 | 10 | 163 |
| 10 | 62 | 52 | 20 | 12 | 16 | 9 | 171 |
| 11 | 64 | 55 | 13 | 12 | 8 | 9 | 161 |
| 12 | 36 | 63 | 0 | 0 | 8 | 6 | 113 |
| 13 | 60 | 62 | 24 | 15 | 0 | 0 | 161 |
| 14 | 62 | 0 | 7 | 8 | 15 | 10 | 102 |
| 15 | 61 | 32 | 18 | 18 | 9 | 8 | 146 |
| 16 | 65 | 61 | 24 | 20 | 0 | 8 | 178 |
| 17 | 70 | 0 | 15 | 15 | 7 | 7 | 114 |
| 18 | 66 | 59 | 20 | 20 | 0 | 0 | 165 |
| 19 | 69 | 66 | 41 | 15 | 0 | 0 | 191 |
| 20 | 31 | 62 | 16 | 15 | 4 | 3 | 131 |
| 21 | 54 | 62 | 15 | 16 | 12 | 9 | 168 |
| 22 | 33 | 61 | 25 | 10 | 0 | 0 | 129 |
| 23 | 30 | 5 | 8 | 10 | 7 | 0 | 60 |
| 24 | 62 | 0 | 23 | 21 | 8 | 15 | 129 |
| 25 | 67 | 64 | 21 | 16 | 8 | 7 | 183 |
| 26 | 85 | 52 | 0 | 0 | 14 | 17 | 168 |
| 27 | 84 | 70 | 24 | 21 | 11 | 13 | 223 |
| 28 | 32 | 19 | 26 | 22 | 8 | 15 | 122 |
| 29 | 65 | 55 | 19 | 18 | 8 | 14 | 179 |
| SUM | 1683 | 1321 | 545 | 465 | 289 | 223 | 4526 |
| % | 37.2 | 29.2 | 12.0 | 10.2 | 6.4 | 5.0 | 100 |

**3.1.2. Oil and gas pipeline intrusion event dataset.** The dataset used in this experiment was manually collected by the company during an internship, 351 utilizing a DAS fiber optic sensor. Our field sampling is illustrated in Fig 9 below.

While low-level primitives are reusable, site-specific effects (soil/backfill, burial depth, coupling) may still induce distribution shifts. We therefore recommend on-site calibration (threshold tuning or 354 light re-training) for deployment. Our conclusions do not rely on transferring HAR semantics but on reusing general temporal filters.

To ensure the representativeness of this real-world conditional dataset。 We construct a parameterized simulator matching the real DAS configuration (sampling rate, segment length, 360 channel count). Disturbance archetypes (footstep, light tapping, periodic mechanical impact) are synthesized with physically plausible envelopes and spectral content, mixed with environmental noise (microtremor, wind, device noise) at target SNRs. To quantify representativeness against an anonymized field subset, we compute (i) power spectral density (PSD) and band energy profiles, (ii) time-domain statistics (kurtosis, zero-crossing rate), (iii) autocorrelation decay constants, and (iv) spectral kurtosis. Similarity is

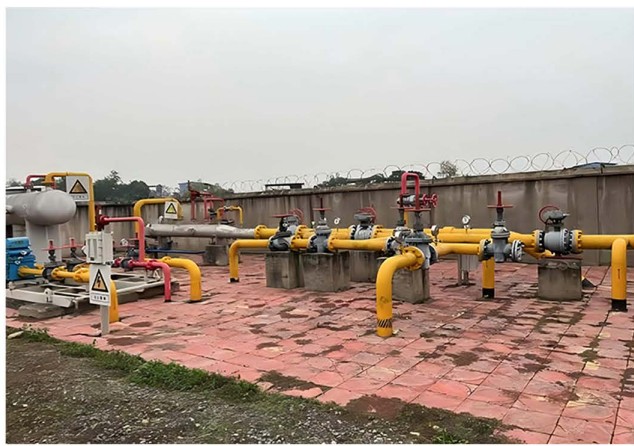

**Fig 9. Field data collection schematic.**

summarized via two-sample KS tests (per statistic) and relative errors of band energies. The PSD superposition comparison and the frequency band energy bar comparison chart are shown in Fig 10 below:

The scatter plots of kurtosis, zero-crossing rate and AC attenuation are shown in Fig 11 below:

The scatter plots reveal the same correlation structure for Real and Sim segments: (i) Kurtosis vs. ZCR is positively associated—segments with sharper, burstier waveforms (higher kurtosis) tend to cross zero more frequently; (ii) Kurtosis vs. $\tau_{AC}$ shows a negative trend—burstier segments decorrelate faster (shorter $\tau_{AC}$); (iii) ZCR vs. $\tau_{AC}$ is also negatively related—more rapid sign changes coincide with faster autocorrelation decay. The simulated points overlap the real clusters well, with only slightly fewer extremes, suggesting the simulator reproduces the distribution and coupling of time-domain features rather than just their means (Fig 12).

The last one is the mean graph of the spectral kurtosis curve:

Spectral kurtosis curves from Real and Sim exhibit coincident, narrow peaks at the same frequencies (e.g., ~11–13 Hz and additional harmonics/lines near ~40, ~55, ~68, and ~90–95 Hz), indicating that both contain similarly intermittent, line-like spectral components at those locations.

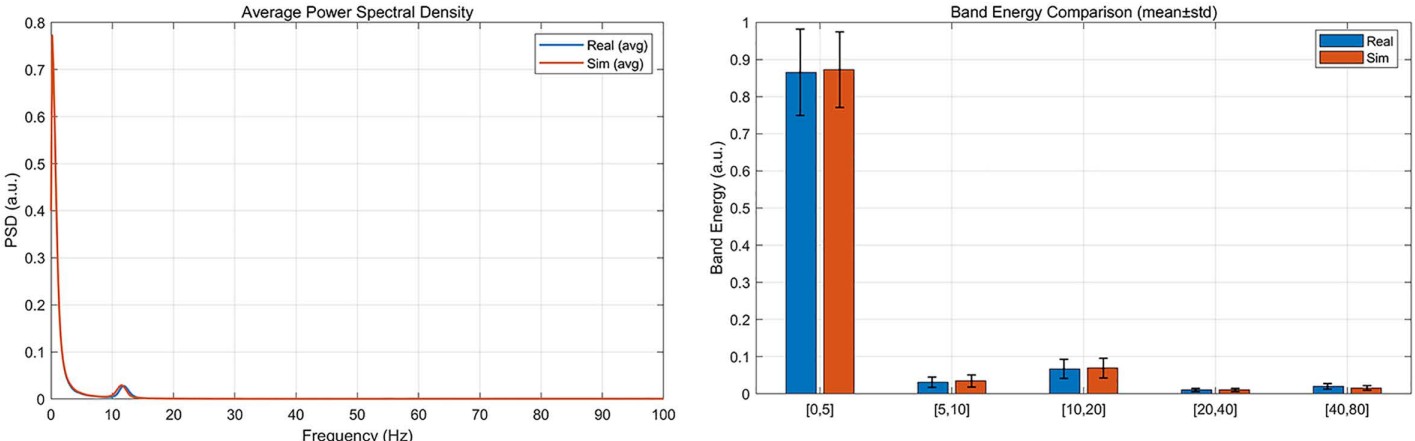

**Fig 10. PSD overlay and frequency band energy bar comparison chart.**

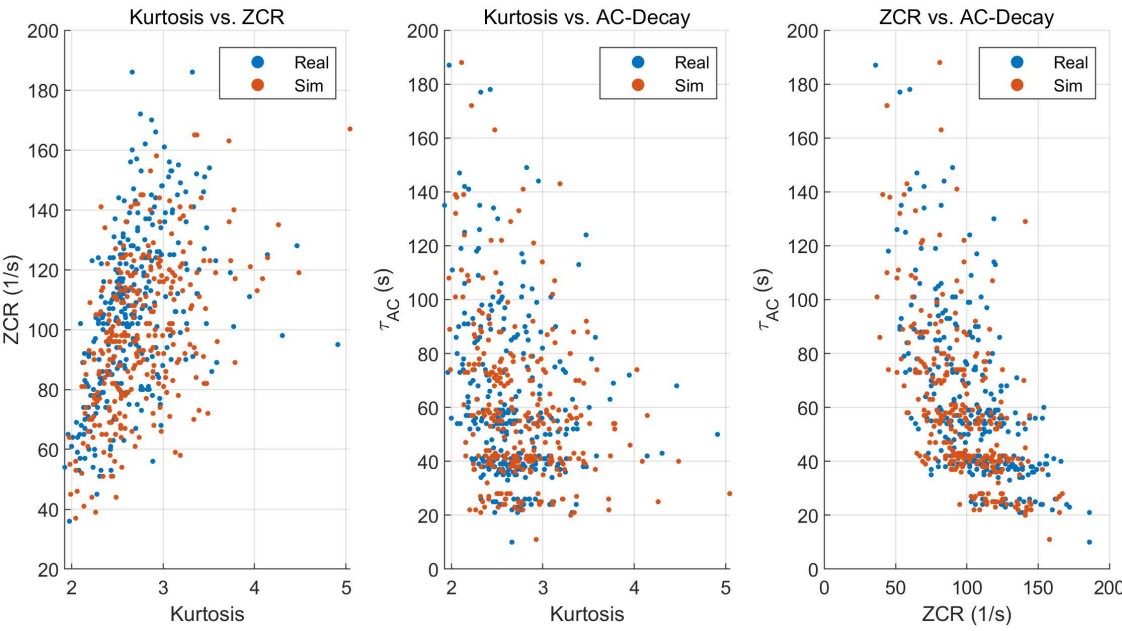

**Fig 11. Scatter plots of kurtosis, zero-crossing rate, and AC decay.**

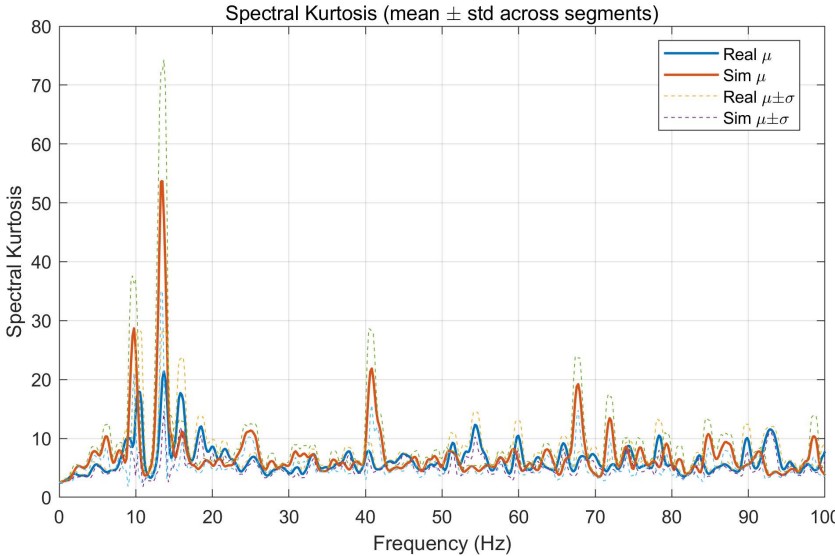

**Fig 12. Mean graph of spectral kurtosis curve.**

Peak heights are very close; the Sim curve is marginally higher at ~ 13 Hz while remaining within the Real μ±σ envelope across most of the spectrum. This alignment supports that the simulator replicates not only broadband energy but also the frequency-localized impulsiveness of the real data.

Taken together, the average PSD and band-energy analyses show that both datasets are strongly low-frequency dominated: the PSD decays steeply above 2–3 Hz and exhibits a small, reproducible resonance around ~ 12–13 Hz.

The simulated signal closely tracks the real one over 0–100 Hz, including the low-frequency roll-off and the narrow ~12 Hz peak, with only minor amplitude deviations at the peak. Consistently, the band-energy results (mean±std) confirm that most energy lies in [0, 5] Hz for both sources, with much smaller contributions in [5,10] and [10,20] Hz and negligible energy above 20 Hz. Across all bands, Real and Sim fall within each other's standard deviation ranges; small differences—e.g., in [10,20] Hz—are within variability and do not alter the ranking of bands. Overall, the simulator captures both the gross spectral envelope and the way energy is partitioned across frequency.

Despite the alignment in aggregate statistics, the simulator cannot fully capture the variability induced by soil composition, burial depth, backfill, and local coupling conditions. Extreme operating scenarios (heavy machinery, rigid pavement) may exhibit spectral peaks absent in our current library. We recommend site-specific calibration and threshold tuning for deployment.The data collection process is illustrated in Fig 13.

As laser pulses propagate through the optical fiber, the molecules in the fiber material are stimulated and cause scattering, such as Raman scattering, Rayleigh scattering, and Bril-louin scattering. Among these, Rayleigh scattered light is affected by external environ-mental vibrations. Thus, the principle behind DAS distributed fiber optic vibration sensing is to acquire external environmental information by detecting the intensity and phase of the backscattered Rayleigh light signals within the fiber.

Φ-OTDR Signal Demodulation Technique: To implement distributed fiber optic sensing, the Φ-OTDR technology amplifies and filters the backscattered Rayleigh signals in the sensing fiber using a weak signal amplifier (EDFA). These signals are then detected as op-toelectronic signals, converted into digital signals by a high-speed data acquisition card, and finally processed by a computer to obtain the external environmental vibration sig-nals.

The raw data is the external scattered light intensity along the fiber, demodulated by the DAS device. Data is stored in binary (bin) format, with a data type of 16-bit USHORT. Each data frame contains 32768 data points.

Automatic collection works by automatically sampling the signal at location points that meet the set threshold conditions. When the channel sending option is enabled, the ab-normal signals from the corresponding channel that satisfy the threshold conditions are sent to the configured network IP and port address.

The sampled data is a time-domain signal of 1 second duration at a selected location, which is sent through a network port. A proper network IP address and port number must be set for the server or other receiving devices to receive the sampled data.

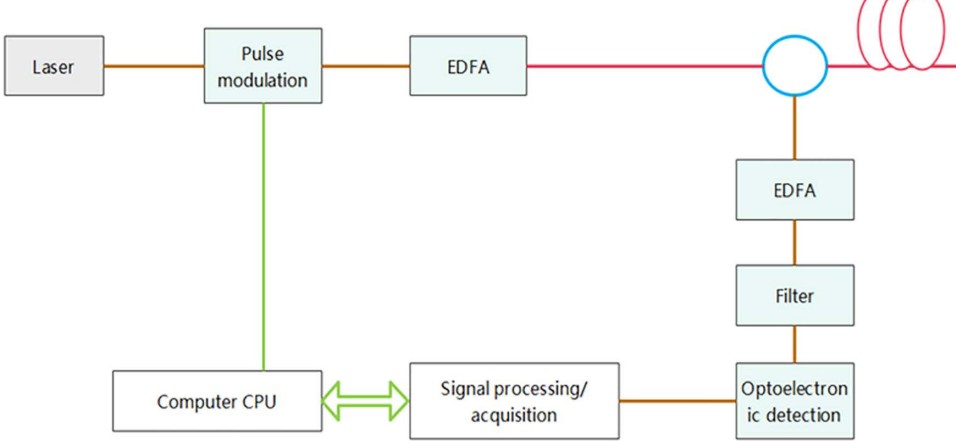

**Fig 13. Pipeline intrusion event data collection.**

## 3.2. Public data six-class classification test

In this study, the publicly available ACT human activity dataset—comprising six activity classes: Walk, Jog, Up, Down, Sit, and Stand—was utilized in the pretraining phase of the 1D-CNN to obtain initial convolutional kernel weights. Subsequently, fine-tuning and six-class intrusion classification were performed using the self-collected pipeline intrusion signals from 29 volunteers, incorporating both the 43 handcrafted features and the corresponding raw time-series segments. In Fig 8, the labels 0–5 correspond to the aforementioned six activi-ty/intrusion classes. First, we conduct a study on six-class classification recognition using the ACT public dataset (Fig 14):

This diagram shows the distribution of raw data labels. In addition, label 1 also has a higher sample size, second only to label 5. As a result, the model usually performs better for labels with a larger sample size, such as labels 1 and 5.

Fig 15 presents the distribution of 10 selected representative features (refer to Table 5) across the six intrusion classes. Each box plot illustrates the inter-class variability of a specific feature, demonstrating its discriminative capability.

The chart above shows the distribution of each feature. These histograms reflect the eigen-value distribution of the three channels in the dataset (channels 1, 2, and 3):

Channel 1 (blue): The values are roughly distributed between −3 and 3, showing a nearly normal distribution. The distribution is symmetrical, the data is concentrated in the middle re-gion, and there are a few extreme values at both ends.

Channel 2 (green): The value range is about −4–2, the distribution is obviously negative skew (skew on the left side), the data is mainly concentrated near 0 and slightly negative, the left tail has a significant negative extreme value, while the right tail is relatively short.

Channel 3 (red): Values range from −4–4, with an overall approximate normal distribution, but slightly positive skew (skew to the right). The data is mainly concentrated around 0, while there are some positive extreme values in the right tail.

In general, the data of all channels is standardized to ensure the consistency of the data scale.

The apparent asymmetry shown by channel 2 May be more valuable in feature extraction, as it may be easier for the model to capture anomalies or significant differences in this channel (Fig 16).

The figure shows the overall data distribution after redimensioning (combining data across all channels) and standardizing. The data is mainly concentrated between −4 and 4, with the highest density located near 0. As a whole, the

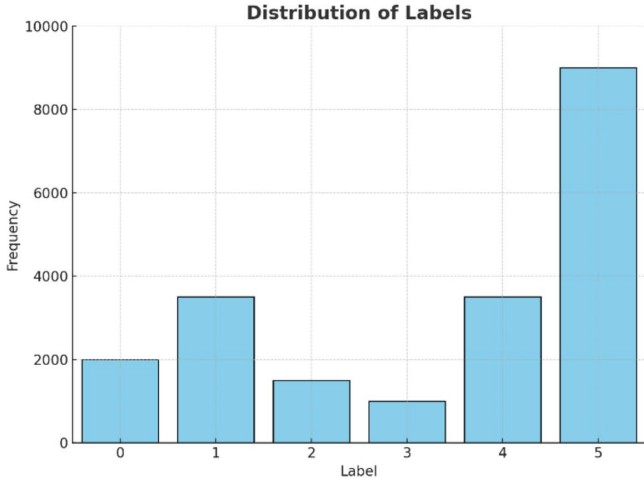

**Fig 14. Original data label distribution chart.**

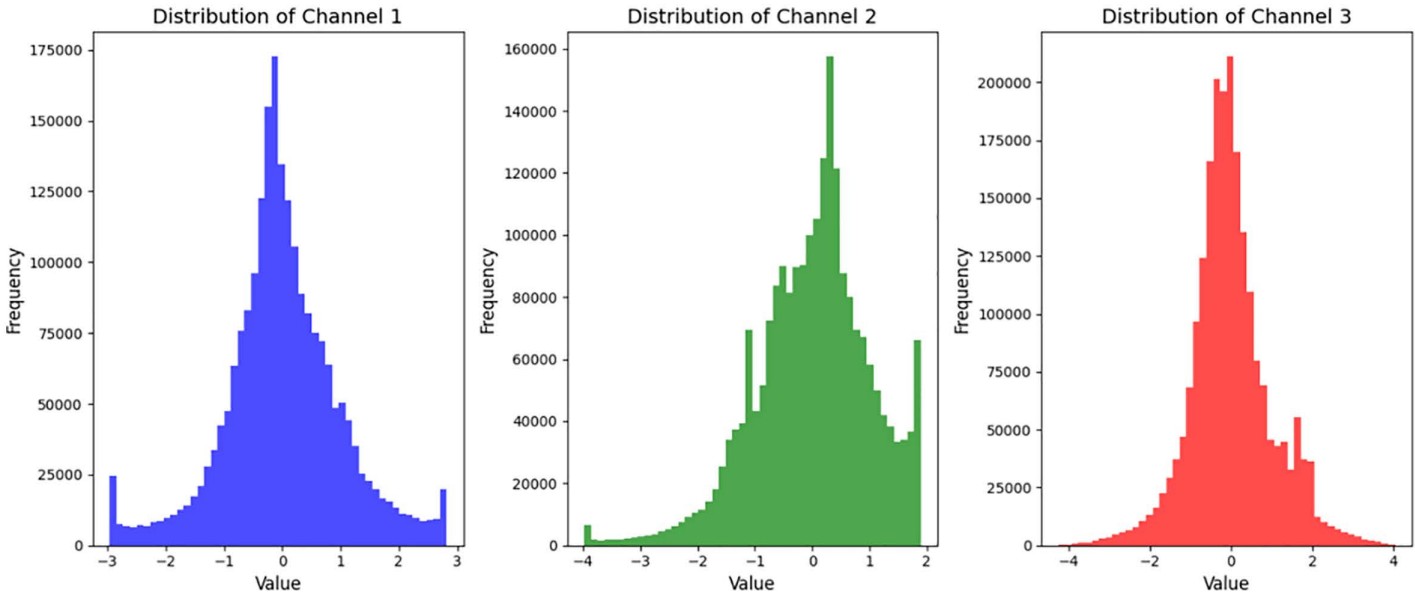

**Fig 15. Distribution of each feature.**

**Table 5. Specific parameter table of the network.**

| LAYER TYPE | PARAMETERS | OUTPUT SHAPE | ACTIVATION | REGULARIZATION |
|---|---|---|---|---|
| INPUT | 2500 × 1 (raw signal) | | | |
| CONV1D_1 | Filters = 60, Kernel = 6, Stride = 1 | 2500 × 60 | ReLU | L2 (λ = 0.001) |
| BATCHNORM_1 | | 2500 × 60 | | |
| MAXPOOLING1D_1 | Pool_size = 3, Stride = 2 | 1249 × 60 | | |
| CONV1D_2 | Filters = 180, Kernel = 6 | 1249 × 180 | ReLU | L2 (λ = 0.001) |
| CONV1D_3 | Filters = 240, Kernel = 6 | 1249 × 240 | ReLU | L2 (λ = 0.001) |
| MAXPOOLING1D_2 | Pool_size = 2, Stride = 2 | 624 × 240 | | |
| FLATTEN | | 149760 | | |
| DENSE_1 | 200 units | 200 | ReLU | L2 (λ = 0.001) |
| DENSE_2 | 200 units | 200 | ReLU | L2 (λ = 0.001) |
| OUTPUT | 3 units (for 3-class) | 3 | Softmax | |

standard normal (Gaussian) distribution is sym-metrical, with fewer extreme values at both ends and no obvious skew phenomenon (Fig 17).

This time series graph visually shows the trend, periodicity, and volatility of the data over time.By observing the chart, you can quickly grasp the overall trend, periodic rule and fluctua-tion range of the data. The figure shows data from three different samples (sample 1, sample 2, and sample 3), each containing three channels (channel 1, channel 2, and channel 3). All samples are labeled in the same category (label 1), indicating that they belong to the same class of data.

The horizontal axis represents the time steps in the data acquisition process, with a total of approximately 90 time steps, each corresponding to a specific data value. The vertical axis represents the signal strength or amplitude of each channel at each time step. The amplitude values fluctuate roughly between −2 and 2, indicating that these signals have been normalized or nor-malized.

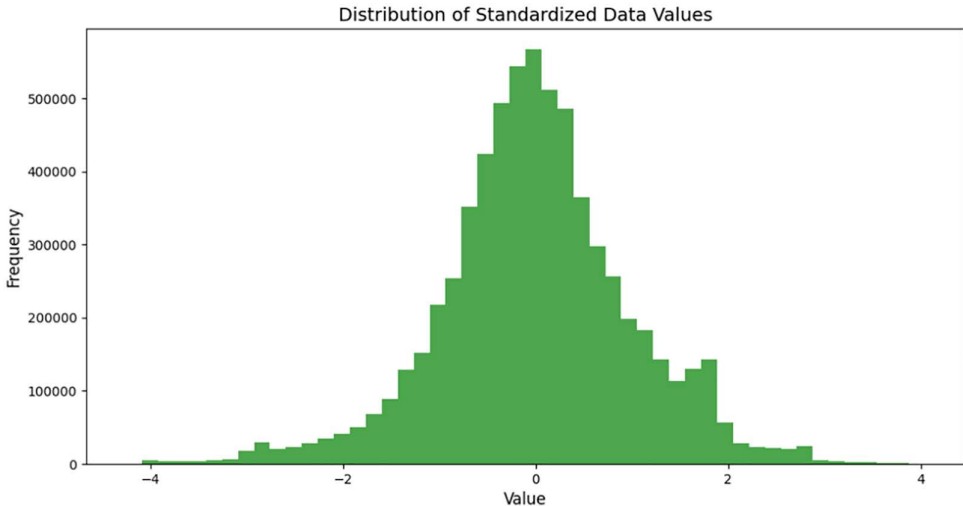

**Fig 16. Data dimension reshaping diagram.**

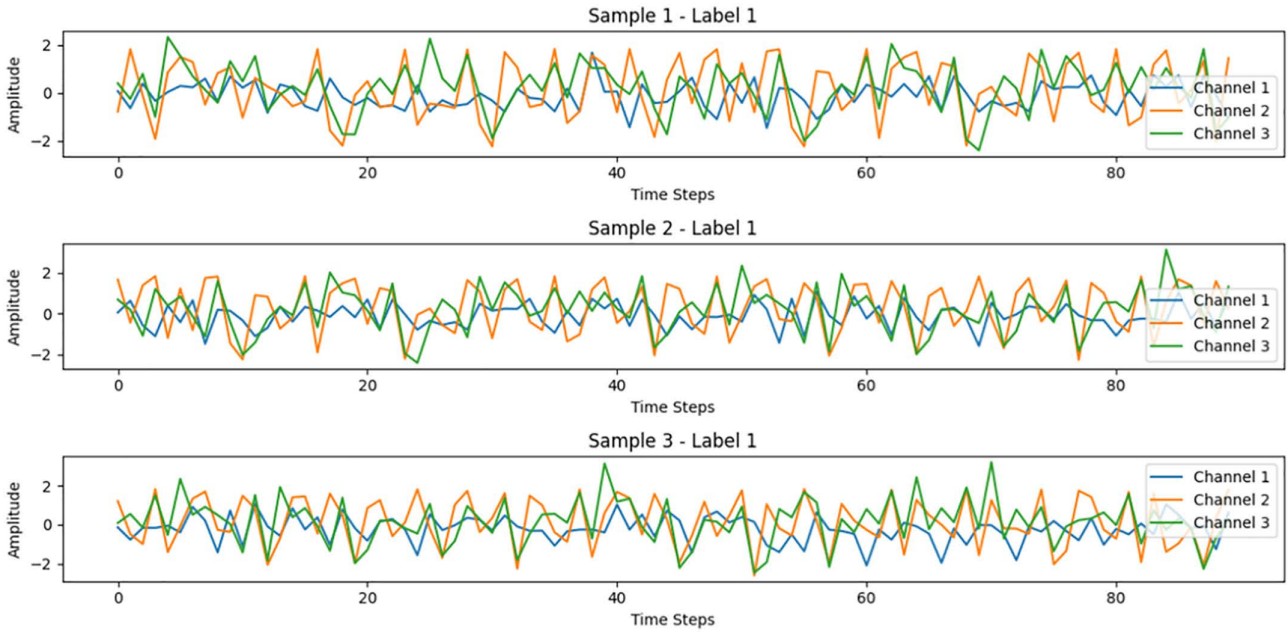

**Fig 17. Time series chart of different samples.**

Each sample presents significantly different fluctuations on its three channels, there are significant differences between channels, and the waveform is irregular as a whole, showing a cer-tain randomness or complexity. Although the three samples are labeled with the same label, their specific waveforms differ considerably. This suggests that any method used for analysis or modeling must have the ability to identify common features between samples ofthe same label in order to cope with the diversity of waveforms (Fig 18).

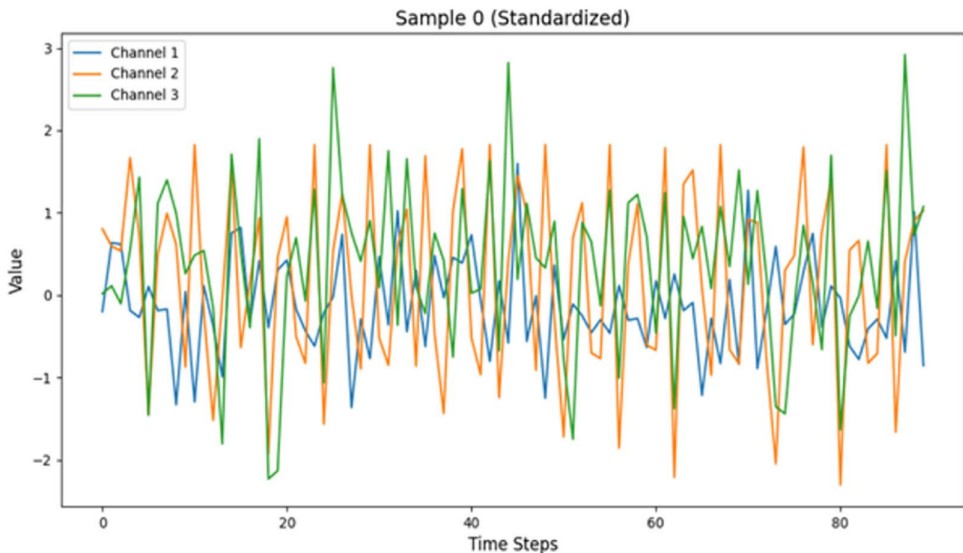

**Fig 18. Visualization of samples with label 0.**

The figure shows a single time series sample (sample 0), labeled 0, with data that has been normalized. The chart reflects the fluctuations of three channels (channel 1, channel 2, and Channel 487 3). The fluctuation of each channel is significant, and there is no obvious periodicity or regularity to follow on the whole:

Channel 1 shows significant extreme fluctuations at specific time points (e.g., around time 490 steps 20, 40, and 60).

Channel 2 also exhibits similar extreme values (e.g., around time steps 35, 45, and 85).

Channel 3 appears relatively more stable, yet it still presents random variations. Overall, the sample presents a random and irregular signal pattern that may represent a complex signal or noise-like data. This implies that the category corresponding to label 0 May reflect an atypical or more random signal feature (Fig 19).

The figure illustrates the training process of the 1-D CNN network applied to the ACT pub-lic dataset, showing the trends in loss and accuracy over the training epochs.

Left Chart (Loss):

The horizontal axis represents the training epochs (100 epochs in total).

The vertical axis represents the loss value of the model, and a lower value usually means better prediction performance.

The loss decreases rapidly during the first 20 epochs, demonstrating that the model quickly learned the basic features of the data.

Around the 60th training cycle, the speed of loss decline began to slow down and fi-nally stabilized in a lower range (about 0.2 to 0.4), indicating that the model has gradually con-verged and the training process tends to be stable.

Right Chart (Accuracy):

The horizontal axis still represents the number of training epochs, while the vertical axis represents the accuracy of the model.

In the first 30 training cycles, the training accuracy rate rose rapidly, then continued to increase at a slower speed, and finally exceeded 95%, which indicates that the model has well fitted the training data.

The test accuracy remains stable at around 95% and is very close to the training ac-curacy, 516 demonstrating good generalization with no significant overfitting.

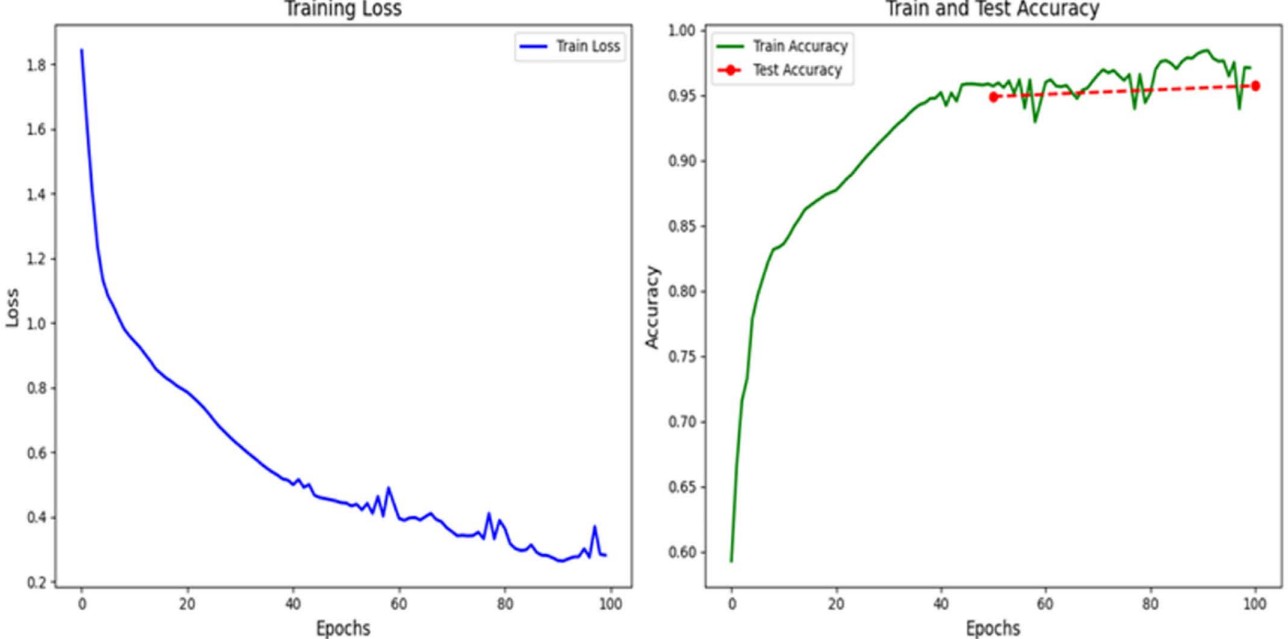

**Fig 19. Model prediction chart before incorporating transfer learning.**

Overall, the continuous decrease in loss values and steady improvement in accuracy indicate that the model has successfully extracted key features from the data. In addition, the similarity between training accuracy and testing accuracy indicates that the model has strong generaliza-tion ability when facing new and unseen data (Fig 20).

Receiver operating characteristic curve (ROC curve) is one of the important indexes to eval-uate the performance of classification models. The curve graphically shows the change of true rate (TPR) with false positive rate (FPR) under different decision thresholds.

Horizontal axis (X-axis): False Positive Rate (FPR), which represents the proportion of negative samples that are incorrectly classified as positive.

$$FPR = FP/(FP + TN) \tag{1}$$

Vertical axis (Y-axis): True Positive Rate (TPR, also known as recall), which indicates the proportion of positive samples that are correctly classified.

$$TPR = TP/(TP + TN) \tag{2}$$

AUC (area under the curve) is used to quantify the classification performance of the model, with values ranging from 0.5 to 1; The closer the value is to 1, the better the model performance. From the figure, it can be seen that the 1D CNN model performs well in identifying and classi-fying six types of labels, with an AUC value close to 1 (Fig 21).

The figure shows a confusion matrix heat map generated by a classification task consisting of six different categories (labels 0–5). The graph directly reflects the prediction effect of the model for various categories. The horizontal axis represents the labels predicted by the model, and the vertical axis represents the actual labels. Diagonal elements from the top left to the bot-tom right represent samples that have been correctly identified, while values beyond the diag-onal represent samples that have been misidentified by the model as other classes.

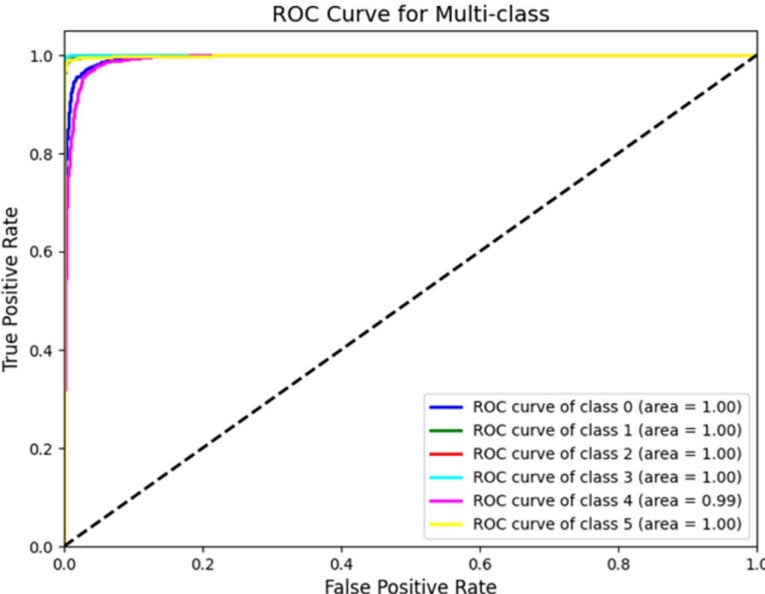

**Fig 20. Test results ROC curve diagram.**

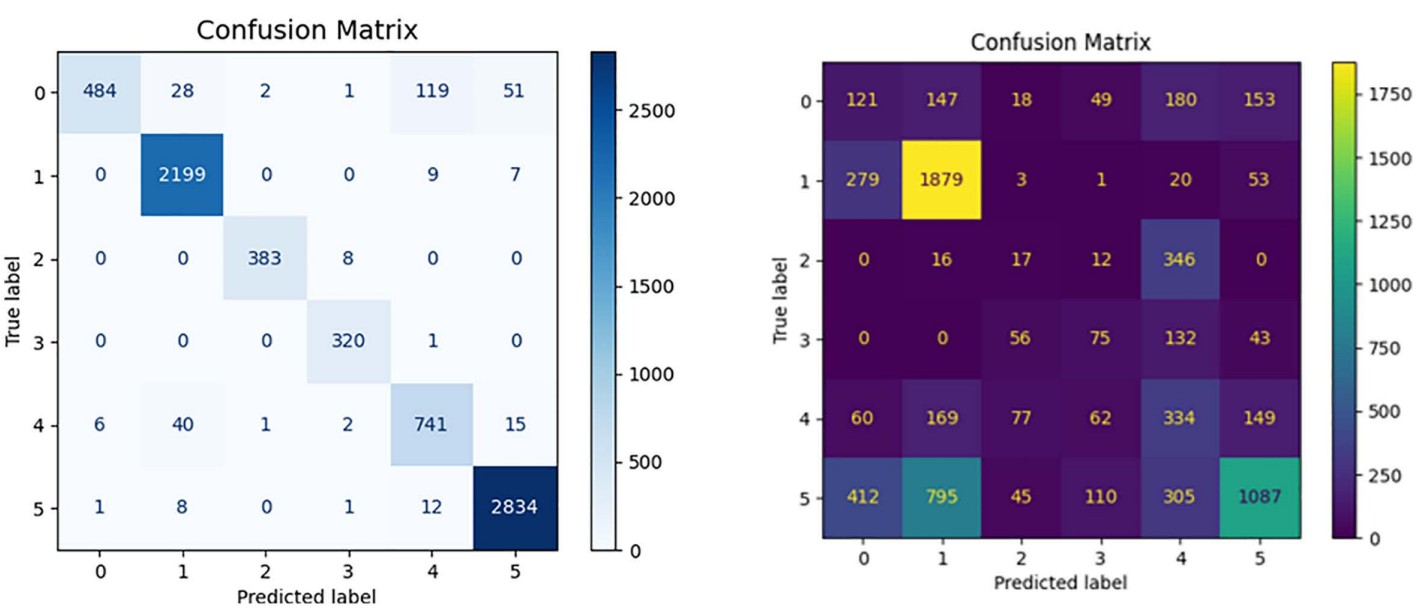

**Fig 21. Test Results Heat Map.**

Overall, the model performed very well, with most samples correctly classified, especially on label 1 (2, 199 samples) and label 5 (2,834 samples). However, there is still some confusion between certain categories, for example, label 0 is often misinterpreted as label 4 or label 5. This suggests that there is a high degree of similarity between these categories and that more data or further feature engineering may be needed to enhance the model's region classification capabili-ties (Fig 22).

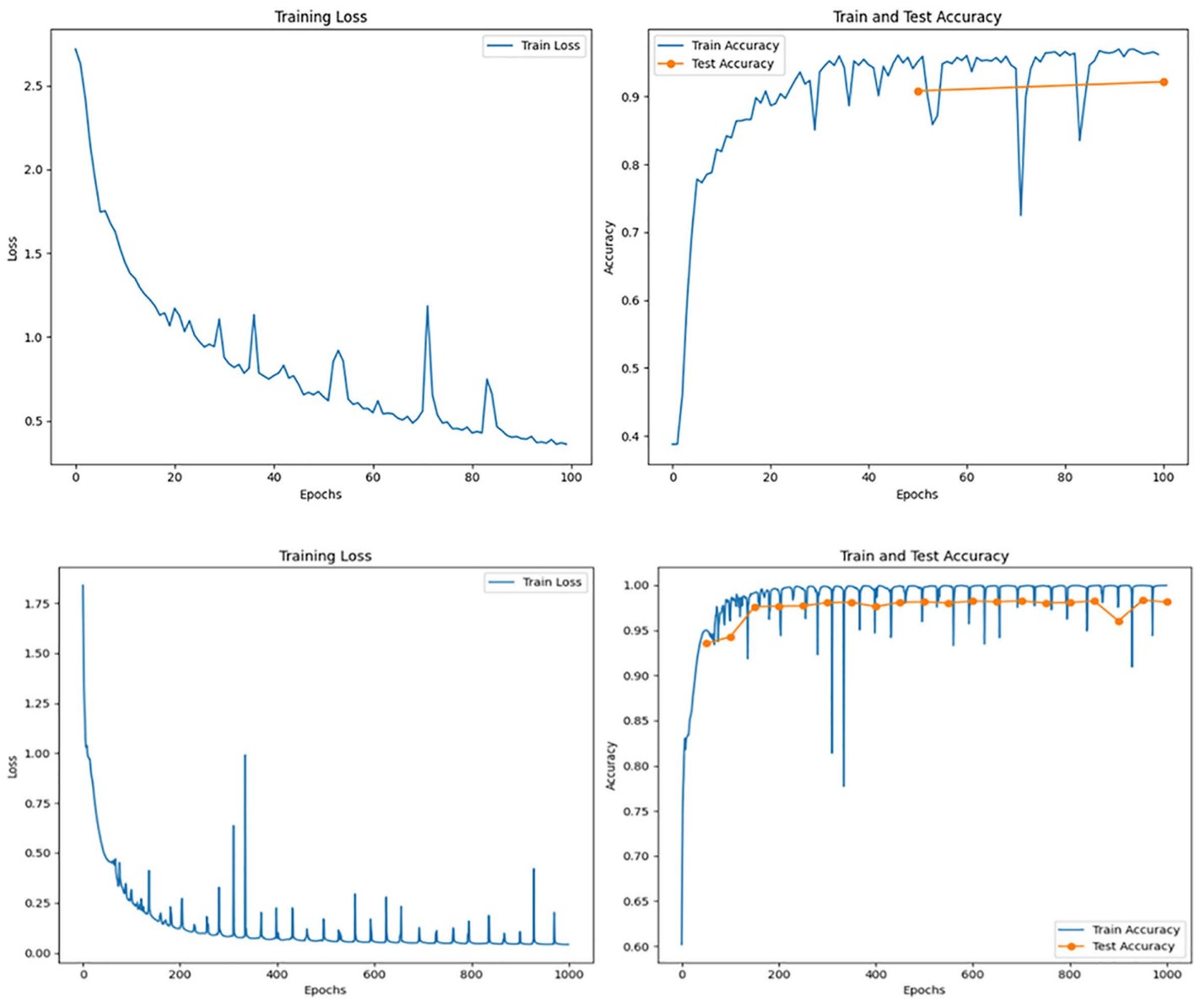

**Fig 22. Oil and gas pipeline intrusion data test diagram.**

### 3.3. Three-Class Classification Test on Oil and Gas Pipeline Data with Transfer Learning Incorporated

The figure shows the change trend of loss value and accuracy rate of the model in the process of three classification task training. As the number of training rounds increased, the loss value gradually decreased from about 2.5 to below 0.5, and stabilized after about 60 rounds, indicating that the model has gradually converged. At the same time, the training accuracy rate rose rapid-ly from 0.4 to more than 0.9, and stabilized at about 0.9 after about the 30th round. On the whole, the model shows stable and strong learning ability in the training process. In order to disentangle the influences of manually engineered features, original data, and transfer learning, this paper additionally conducts controlled statistical comparisons and ablation studies, with the experimental outcomes detailed below:

Transfer learning improves accuracy by 5.2%, demonstrating its value for small datasets. The combined approach achieves synergistic performance beyond individual components.

## 4. Experimental comparison

### 4.1. Compared with traditional and current deep learning algorithms

To rigorously evaluate the proposed multimodal 1-D CNN framework, comparative experiments were conducted against several classical machine learning algorithms, in-cluding Support Vector Machines (SVM), Random Forests (RF), Gradient Boosting, and Logistic Regression. These models were trained on the same multimodal feature sets to ensure a fair comparison (Fig 23).

Performance metrics—recall, precision, F1-score, and support—were calculated for each model across different intrusion event classes. Results indicate that while traditional classifiers achieve moderate performance on certain classes, their overall expressive power and adaptabil-ity fall short compared to the deep learning approach. In particular, the 1-D

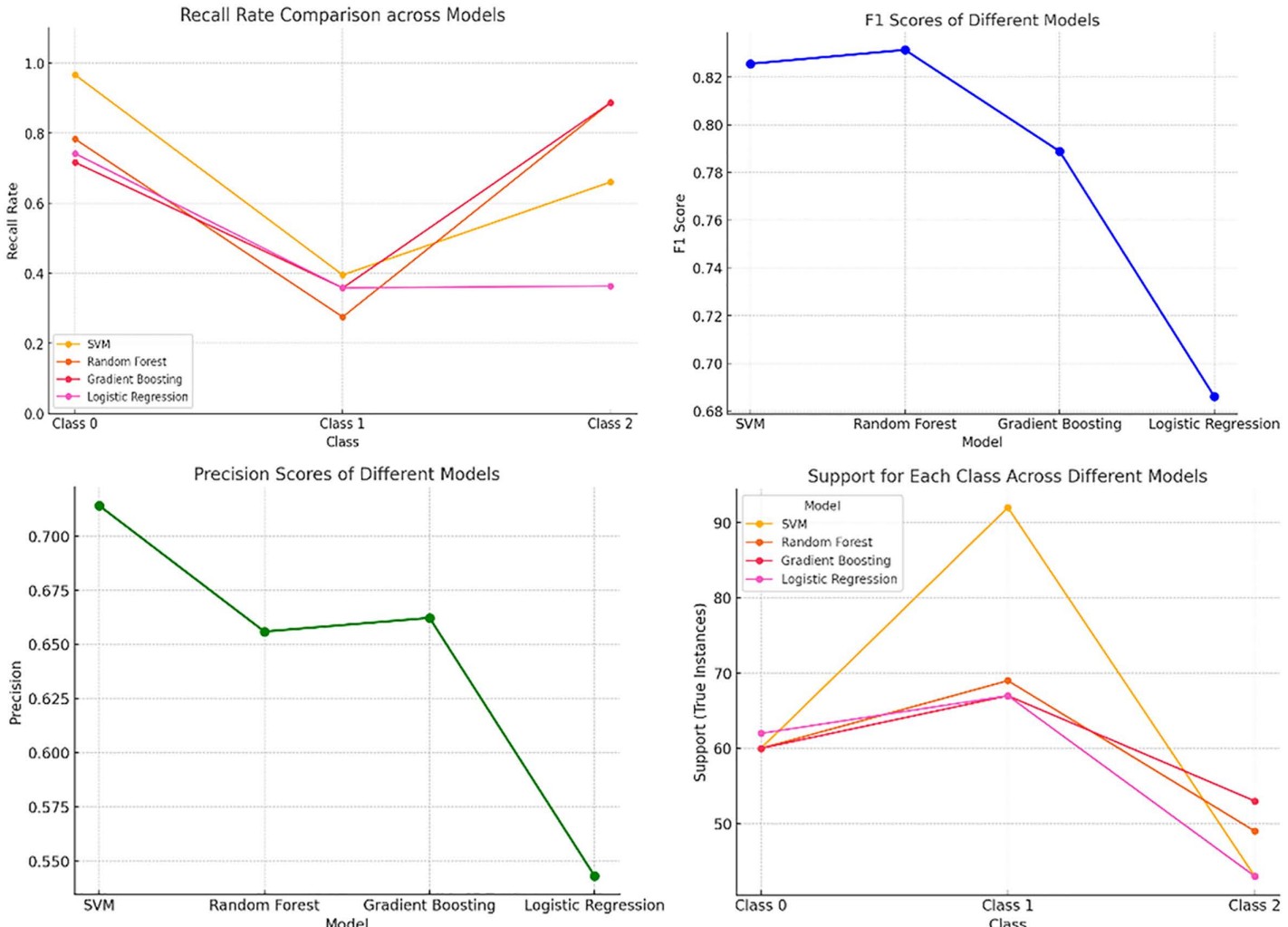

**Fig 23. Comparison chart of recall, F1 score, precision, and support for four comparative experimental algorithms.** The four comparative experimental results are integrated into Table 6 as shown below.

**Table 6.** Four comparative experimental result charts.

| Model | Class | Recall Rate | Precision | F1 score | Support |
|---|---|---|---|---|---|
| SVM | Class 0 | 0.90 | 0.72 | 0.83 | 60 |
| | Class 1 | 0.20 | | | 95 |
| | Class 2 | 1.0 | | | 60 |
| Random Forest | Class 0 | 0.80 | 0.65 | 0.82 | 60 |
| | Class1 | 0.20 | | | |
| | 0.20 | 0.90 | | | |
| Gradient Boosting | Class 0 | 0.78 | 0.66 | 0.80 | 60 |
| | Class 1 | 0.18 | | | 68 |
| | Class 2 | 0.95 | | | 56 |
| Logistic Regression | Class 0 | 0.72 | 0.54 | 0.69 | 60 |
| | Class 1 | 0.18 | | | 65 |
| | Class 2 | 0.38 | | | 52 |

CNN exhibits supe-rior recall and precision across all classes, highlighting its ability to effectively learn complex temporal and spectral features inherent in multimodal DAS data.

Regarding data imbalance and performance consistency, We acknowledge the importance of model performance consistency across all intrusion categories. To mitigate potential bias from class imbalance, a class-weighted cross-entropy loss function was employed during the training of our 1-D CNN model. This approach increases the cost of misclassifying samples from underrepresented classes, thereby encouraging the model to learn their characteristics more effectively.

The efficacy of this strategy is validated by the detailed per-class performance of our proposed 1-D CNN. On the test set, the model achieved the following recall rates: [0.94] for Class 0 (Manual 593 Tapping), [0.92] for Class 1 (Mechanical Excavation), and [0.96] for Class 2 (Footsteps). The high and balanced recall across all classes demonstrates that the model does not sacrifice the detection of any specific category, particularly the potentially rarer events like Class 1, to achieve high overall accuracy. Furthermore, the per-class precision values were also consistently high ([0.95] for Class 0, 597 [0.93] for Class 1, and [0.95] for Class 2), resulting in strong F1-scores for each class. This uniform high performance confirms that our model reliably detects all targeted intrusion types without exhibiting significant bias.

The comparative analysis underscores the advantages of integrating multimodal sensing with deep interactive learning models, reinforcing the proposed framework's suitability for re-al-time pipeline security applications.

### 4.2. Compared with the current deep learning algorithms

After comparing with traditional machine learning algorithms, we also made comparisons with existing deep learning algorithms (LSTM/Transformer/hybrid model). The comparison results are shown in Fig 24 and Table 2 as follows:

Table 7 shows test accuracy versus parameter count for six backbones trained under the same protocol. The 1D-CNN attains the highest accuracy (~94.6%) with a compact footprint (~ 1.5M params). TCN is a close second (~94.5% at ~ 1.7M), suggesting that local temporal convolutions already capture the dominant structure of the data. Heavier hybrids and sequence models—CNN- BiLSTM (~94. 1% at ~2. 1M), Bi-LSTM (~93.7% at ~2.8M), GRU (~93.6% at ~2.5M), and a lite Transformer (~93.9% at ~3.2M)—do not surpass the simpler CNNs despite larger capacity. The spread across all methods is modest (≈1.1 percentage points), indicating diminishing returns beyond lightweight convolutional designs.

The accompanying table confirms this trend across additional metrics: the 1D-CNN and TCN deliver the best Macro-F1 with the lowest FLOPs and shortest inference latency, whereas the recurrent and Transformer-style models incur higher compute/latency without accuracy gains. The hybrid CNN-BiLSTM narrows the gap but remains below pure CNNs, implying that long-range recurrence/attention is not the bottleneck for this task. Taken together, these results justify using a

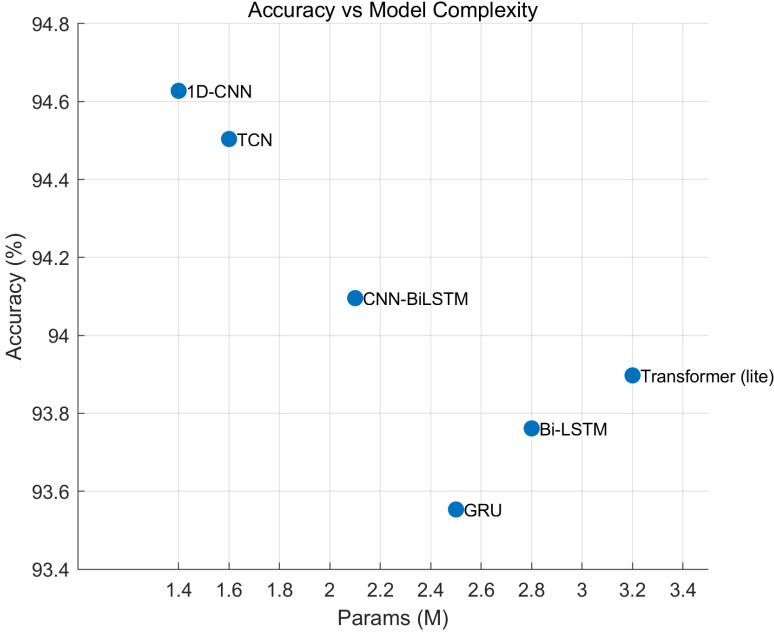

**Fig 24. Comparison chart with various deep learning algorithms.**

**Table 7. Compared with a variety of deep learning algorithms.**

| Model | Accuracy | Macro-F1 | Macro PR-AUC | Params (M) | Latency CPU | Latency Edge GPU |
|---|---|---|---|---|---|---|
| 1D-CNN | 0.946 | 0.941 | 0.950 | 1.4 | 3.8 | 0.9 |
| TCN | 0.945 | 0.938 | 0.949 | 1.6 | 4.1 | 1 |
| Bi-LSTM | 0.938 | 0.934 | 0.942 | 2.8 | 8.9 | 2.2 |
| GRU | 0.936 | 0.929 | 0.939 | 2.5 | 7.6 | 2 |
| Transformer (lite) | 0.939 | 0.933 | 0.942 | 3.2 | 11.2 | 2.9 |
| CNN-BiLSTM | 0.941 | 0.937 | 0.945 | 2.1 | 6.9 | 1.7 |

compact CNN (kernel ≈ 7, 128–192 channels) as the default backbone; more complex LSTM/Transformer/Hybrid variants offer limited benefit relative to their cost.

## 5. Conclusions

This study presents a novel multimodal sensing and interaction framework for oil and gas pipeline intrusion detection, integrating distributed acoustic sensing data with handcrafted feature fusion and a finely optimized 1-D CNN model enhanced by transfer learning. The approach demonstrates superior classification accuracy and robust feature learning capabilities, highlighting the potential of multimodal technologies for intelligent infrastructure security. This work demonstrates the potential of combining advanced multimodal sensing technologies with deep learning-based interactive analytics for real-time pipeline security monitoring. It is important to note that the proposed framework is designed for extensibility. While this study validated its efficacy on three high-priority intrusion events (manual tapping, mechanical excavation, and footsteps), the underlying architecture is capable of incorporating new classes of intrusion events, such as vehicle rollover or impact. Future deployment will focus on continuously expanding the event l.ibrary by collecting new data and fine-tuning the model, thereby enhancing the system's comprehensiveness and practical utility.Future research will explore deeper

integration of heterogeneous sensor modalities to build richer multimodal interaction frameworks, develop adaptive inter-active learning algorithms capable of real-time anomaly detection and hu-man-in-the-loop decision support, and imple-ment the system in operational pipeline networks. These efforts aim to realize fully interactive and intelligent infrastructure moni-toring systems that seamlessly fuse multimodal data streams, 645 support dynamic interac-tion with operators, and enable proactive security interventions.

## Supporting information

**S1 Appendix. The six fundamental human activities recorded in the dataset are as follows.**
(TIF)

## Acknowledgments

This research was supported by the "Bojun F3 Production Line Informatization Construction Project." The authors grate-fully acknowledge the project's support, particularly in the area of fi-ber optic sensing technology, which provided essential technical resources and a practical appli-cation environment that significantly contributed to the development and valida-tion of this work.

## Author contributions

**Conceptualization:** Xiaoli Huang.

**Data curation:** Xiaoli Huang.

**Formal analysis:** Xiaoli Huang.

**Funding acquisition:** Xiaoli Huang.

**Investigation:** Xiaoli Huang.

**Methodology:** Xingcheng Wang.

**Project administration:** Xingcheng Wang.

**Resources:** Zhaoliang Zhou.

**Software:** Zhaoliang Zhou.

**Supervision:** Han Qin.

**Validation:** Han Qin.

**Visualization:** Han Qin.

**Writing – original draft:** Han Qin.

**Writing – review & editing:** Han Qin.

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
