## [Decision Letter · Decision Letter 0]

13 Oct 2025

Identification and Classification of Oil and Gas Pipeline Intru-sion Events Based on 1-D CNN Network

PLOS ONE

Dear Dr. Huang,

Thank you for submitting your manuscript to PLOS ONE. After careful consideration, we feel that it has merit but does not fully meet PLOS ONE’s publication criteria as it currently stands. Therefore, we invite you to submit a revised version of the manuscript that addresses the points raised during the review process.

The manuscript has been evaluated by two reviewers, and their comments are available below.

The reviewers have raised a number of concerns that need attention. In particular, they request additional information on methodological aspects of the study, additional comparisons against more recent deep learning models, and further discussions of the limitations.

Could you please revise the manuscript to carefully address the concerns raised?

We look forward to receiving your revised manuscript.

Kind regards,

Helen Howard

Staff Editor

PLOS ONE

Additional Editor Comments (if provided):

Reviewers' comments:

Reviewer's Responses to Questions

**Comments to the Author**

1. Is the manuscript technically sound, and do the data support the conclusions?

Reviewer #1: Yes

Reviewer #2: Partly

2. Has the statistical analysis been performed appropriately and rigorously?

Reviewer #1: Yes

Reviewer #2: I Don't Know

3. Have the authors made all data underlying the findings in their manuscript fully available?

Reviewer #1: Yes

Reviewer #2: No

4. Is the manuscript presented in an intelligible fashion and written in standard English?

Reviewer #1: Yes

Reviewer #2: Yes

Reviewer #1: 1- The manuscript integrates both handcrafted features and raw DAS signals within a 1-D CNN. Could the authors clarify how feature-level fusion was implemented, and whether alternative fusion strategies were considered?

2 - The paper employs transfer learning from a human activity dataset to improve generalization. How do the authors ensure that the domain gap between human motion data and pipeline intrusion signals does not introduce bias or spurious correlations?

3- The dataset used for pipeline intrusion detection was collected under confidentiality agreements. Although a simulated dataset is provided, how representative is this dataset of real-world conditions, and what limitations might this impose for reproducibility and field deployment?

4- The experiments report high accuracy and robust classification performance. Could the authors expand on how their model handles class imbalance and whether performance is consistent across all intrusion categories, particularly rare events?

5- The architecture of the proposed 1-D CNN was optimized via empirical tuning and grid search. How sensitive are the results to hyperparameter choices, and have the authors explored automated methods such as Bayesian optimization or evolutionary search for architecture refinement?

6- The proposed approach demonstrates superior results compared to classical machine learning baselines. However, how does the method compare against more recent deep learning models, as LSTMs, Transformers, or hybrid CNN-RNN architectures, for time-series intrusion detection tasks?

Reviewer #2: 1.Why is a one-dimensional convolutional neural network adopted? Can other neural networks not be used? The paper should conduct a comparative analysis of the advantages and disadvantages of multiple methods.Some of the latest references are available for reference, such as DOI10.1016/j.jngse.2020.103716;10.1016/j.jngse.2021.104175

2.Pipeline systems are subjected to a multitude of intrusion events, such as excavation, impact, vehicle rollover, and human activities. The article only studies human activities; how to ensure the applicability of its method is a question that needs to be addressed.

3."Reyes-Ortiz, J., Anguita, D., Ghio, A., Oneto, L., & Parra, X. (2013). Human Activity Recognition

236 Using Smartphones [Dataset]. UCI Machine Learning Repository. https://doi.org/10.24432/C54S4K",How do smartphone-based datasets reflect the characteristics of pipeline intrusion events?

**Do you want your identity to be public for this peer review?** For information about this choice, including consent withdrawal, please see our Privacy Policy

Reviewer #1: No

Reviewer #2: No

---

## [Author Response · Author response to Decision Letter 1]

21 Oct 2025

First and foremost, we would like to express our sincere gratitude to the reviewers for their thorough evaluation of our manuscript and for providing insightful comments and constructive suggestions. We have carefully considered all the feedback and have made extensive revisions to the manuscript accordingly. In particular, based on the reviewers' recommendations, we have incorporated additional detailed experiments and expanded the corresponding explanations in the text.

---

## [Decision Letter · Decision Letter 1]

19 Nov 2025

Identification and Classification of Oil and Gas Pipeline Intru-sion Events Based on 1-D CNN Network

PONE-D-25-41494R1

Dear Dr. Huang,

We’re pleased to inform you that your manuscript has been judged scientifically suitable for publication and will be formally accepted for publication once it meets all outstanding technical requirements.

Kind regards,

Muhammad Ahsan, Ph.D.

Academic Editor

PLOS ONE

Additional Editor Comments (optional):

Reviewers' comments:

Reviewer's Responses to Questions

**Comments to the Author**

Reviewer #1: All comments have been addressed

Reviewer #2: All comments have been addressed

Reviewer #3: All comments have been addressed

2. Is the manuscript technically sound, and do the data support the conclusions?

Reviewer #1: Yes

Reviewer #2: Yes

Reviewer #3: Yes

3. Has the statistical analysis been performed appropriately and rigorously?

Reviewer #1: Yes

Reviewer #2: Yes

Reviewer #3: Yes

4. Have the authors made all data underlying the findings in their manuscript fully available?

Reviewer #1: Yes

Reviewer #2: Yes

Reviewer #3: Yes

5. Is the manuscript presented in an intelligible fashion and written in standard English?

Reviewer #1: Yes

Reviewer #2: Yes

Reviewer #3: Yes

Reviewer #1: (No Response)

Reviewer #2: The authors has made a sound response to the reviewer's comments, and I think it can be published .

Reviewer #3: Dear Authors,

Thank you for this detailed revision. The revised manuscript deserves the acceptance of publication.

**Do you want your identity to be public for this peer review?** For information about this choice, including consent withdrawal, please see our Privacy Policy

Reviewer #1: No

Reviewer #2: No

Reviewer #3: **Yes: ** Ghassan Abdul-Majeed

---

## [Editor Report · Acceptance letter]

PONE-D-25-41494R1

PLOS ONE

Dear Dr. Huang,

I'm pleased to inform you that your manuscript has been deemed suitable for publication in PLOS ONE. Congratulations! Your manuscript is now being handed over to our production team.

Kind regards,

on behalf of

Dr. Muhammad Ahsan

Academic Editor

PLOS ONE